# Characteristics and Outcomes of Patients with Cancer Pain Placed in an Emergency Department Observation Unit

**DOI:** 10.3390/cancers14235871

**Published:** 2022-11-29

**Authors:** Pavitra Parimala Krishnamani, Aiham Qdaisat, Monica Kathleen Wattana, Demis N. Lipe, Marcelo Sandoval, Ahmed Elsayem, Maria Teresa Cruz Carreras, Sai-Ching Jim Yeung, Patrick S. Chaftari

**Affiliations:** 1Department of Emergency Medicine, The University of Texas MD Anderson Cancer Center, Houston, TX 77030, USA; 2Department of Emergency Medicine, Baylor College of Medicine, Houston, TX 77030, USA; 3IQVIA Biotech, Morrisville, NC 27560, USA

**Keywords:** pain, cancer, observation unit, clinical decision unit, oncology, emergency medicine

## Abstract

**Simple Summary:**

Pain is an important yet undertreated complication of cancer that has been shown to affect patients’ quality of life. For patients presenting in an acute care setting with cancer-related pain, there have traditionally been two routes to management: inpatient or outpatient. However, with the advent of observation units, there is now an opportunity for these patients to utilize hospital resources without an inpatient stay. To better understand the role of an observation unit in pain management, this study analyzed charts for patients who had their pain managed in an observation unit. Patient characteristics and outcomes were statistically analyzed and summarized. Predictors of admission or discharge from the observation unit were also derived from the analysis. Factors that predicted an inpatient hospitalization from the observation unit included patients’ pain scores and the need for consult service recommendations while in the observation unit. Indeed, this research showed that patients in an observation unit for pain management received appropriate consultations and admissions when necessary. However, most were discharged home safely and without a quick return to the acute care setting.

**Abstract:**

Pain remains an undertreated complication of cancer, with poor pain control decreasing patients’ quality of life. Traditionally, patients presenting to an emergency department with pain have only had two dispositions available to them: hospitalization or discharge. A third emerging healthcare environment, the emergency department observation unit (EDOU), affords patients access to a hospital’s resources without hospitalization. To define the role of an EDOU in the management of cancer pain, we conducted a retrospective study analyzing patients placed in an EDOU with uncontrolled cancer pain for one year. Patient characteristics were summarized using descriptive statistics and predictors of disposition from the EDOU and were identified with univariate and multivariate analyses. Most patients were discharged home, and discharged patients had low 72-hour revisit and 30-day mortality rates. Significant predictors of hospitalization were initial EDOU pain score (odds ratio (OR) = 1.12; 95% CI 1.06–1.19; *p* < 0.001) and supportive care (OR = 2.04; 95% CI 1.37–3.04; *p* < 0.001) or pain service (OR = 2.67; 95% CI 1.63–4.40; *p* < 0.001) consultations. We concluded that an EDOU appears to be the appropriate venue to care for a subsegment of patients presenting to an emergency department with cancer pain, with patients receiving safe care as well as appropriate consultation and admission when indicated.

## 1. Introduction

Pain is a common concern among patients with cancer, and poor pain control is associated with sleep disturbances and mood disorders [1]. The goal of pain control in this population is to optimize patient comfort and function and avoid unnecessary side effects. Despite efforts to achieve this goal, more than half of patients with cancer receive insufficient pain control, and a quarter die in pain, highlighting oligoanalgesia as a continuing problem in cancer care [2]. Inadequate analgesia for cancer-related pain is a demonstrated concern across healthcare settings, as cancer patients frequently present with pain to emergency departments (EDs) in the United States [3,4,5,6]. 

Recommended strategies to improve pain management in patients with cancer have included more frequent reassessments, earlier initiation of pain control with morphine, and appropriate dosing of morphine when administered [1,7,8,9]. However, appropriate analgesia and close monitoring require more time than we would expect patients to spend in an ED but may not require the full length of time associated with an inpatient hospitalization. 

Observation medicine is an emerging mode of healthcare delivery that can offer patients with cancer pain access to a hospital’s pain management solutions and specialists without an inpatient hospitalization. In cancer-related care, observation medicine is novel; only two ED observation units (EDOUs) have been established in the United States specifically in the service of cancer patients. Previously, however, protocol-driven general EDOUs have been found to be successful in managing other types of pain, specifically chest pain and abdominal pain [3,10,11]. Developing specific protocols for cancer-related pain management in an EDOU requires an understanding of the characteristics that patients with cancer-related pain in this setting possess. This is a domain that has not yet been explored in the literature to our knowledge. This study aims to better define the characteristics and outcomes of cancer patients placed in an EDOU with cancer-related pain.

## 2. Materials and Methods

A single-center retrospective study was conducted at The University of Texas MD Anderson Cancer Center, Houston, Texas, and included all patients who were placed in the EDOU for pain after visiting the ED between 1 March 2019 and 29 February 2020. Patients were placed in the EDOU during the study’s timeframe only when medically necessary. Patients were placed in observation as opposed to hospitalized inpatient based on the two-midnight rule for Medicare patients and Millman care guidelines for managed care or commercial payers. Screening using these criteria was achieved at the time of treating physician documentation with the assistance of case managers, as was standard during the study period at The University of Texas MD Anderson Cancer Center. Primary oncologists were notified about all placements in the EDOU and, thereby, had an opportunity to change their patients’ plan of care. When the EDOU no longer had bed capacity, patients were placed in observation on their primary service—these patients were not included in this study’s analysis.

Excluded were patients aged <18 years, non-cancer patients, patients placed on observation under a primary service outside of the EDOU, and patients for who pain was not the primary reason for observation. Data, including patients’ demographics, clinical and cancer-related data, pharmacy data, and EDOU-related data, were collected from the institution’s data warehouse using Epic SlicerDicer and manual review.

Pain was identified as the primary reason for observation based on treating physician documentation. Pain scores were documented on nursing assessments using a verbal rating scale between 0 and 10, where 0 represents an absence of pain and 10 represents the worst pain a patient can imagine. Analyzed data included scores obtained on presentation to the ED and EDOU placement as well as during a patient’s stay in these care areas. The primary outcomes of interest were various patient characteristics and ultimate disposition from the EDOU. We also compared 30-day mortality and 72-hour ED revisit rates for those admitted and discharged from the EDOU. Patients were considered to be established with our supportive care and/or pain service if they had completed at least one appointment with either of those services within 90 days prior to the indexed ED presentation.

Patients’ characteristics were summarized using descriptive statistics: counts and percentages were derived for categorical variables; means, standard deviations or medians, and interquartile ranges (IQR) were derived as appropriate for continuous variables. Statistical inference was used to compare different groups, with the chi-squared test or the Fisher exact test used in the analysis of categorical variables as appropriate. Mann–Whitney U tests were used in the analysis of continuous variables, as all the continuous variables failed the normality test. Univariate logistic regression analysis was performed to determine the association between each clinical variable and hospital admission. Relevant variables and statistically significant clinical variables from the univariate analysis were further analyzed using a multiple logistic regression model reporting the odds ratio (OR) with 95% confidence interval (95% CI).

The institutional review board of The University of Texas MD Anderson Cancer Center approved this study and granted waivers of informed consent (Protocol DR08-0666). All statistical analyses were performed using R software for Windows (The R Foundation, http://www.r-project.org, version 4.0.3, accessed on 2 February 2021).

## 3. Results

During the study period, there were 28,358 patient visits to our ED. Of the 10,458 patients who had pain as a presenting symptom, 1420 (13.6%) were eventually placed in the EDOU. After application of further exclusion criteria (including pain not being the primary reason for observation), 668 (47.0%) out of the 1420 visits were identified as eligible for analysis (Figure 1). For the purpose of the analysis, and to reduce inaccuracy related to multiple visits from the same patient during the study period, only the first visit for each patient was included, resulting in 600 unique visits being included for analysis (Figure 1).

The median age of the final cohort was 59 years (IQR 48–68 years), and the majority were either White (58.5%) or Black (20.0%). Most of the patients in the cohort were women (60.8%). Breast, colorectal, and lung cancer were the most common cancer types, and a few patients (2.7%) had more than one cancer type (Table 1).

Almost half of the patients had a history of hypertension (46.8%). Other frequent comorbidities were diabetes (24.0%) and renal failure (21.0%). Chemotherapy within 30 days prior to the ED presentation was identified in 198 (30.0%) patients, and 89 (13.2%) visits were for patients who had had immunotherapy within 1 year prior to presentation. Almost one-third (30.2%) of the visits analyzed were for patients who were actively receiving radiotherapy, defined as radiotherapy within 30 days prior to the ED presentation. Only 96 (16.0%) and 97 (16.2%) of the patients were found to be established with pain and supportive care services, respectively. In addition, opioids were prescribed within 45 days prior to the ED presentation in more than half of the patients (56.5%).

The abdomen (29.0%), back (17.3%), and extremities (15.8%) were the most common locations of identified pain (Table 2). Chest pain was reported as the reason for observation in 55 (9.2%) visits. The median initial pain score recorded upon presentation to the ED was 7 on the verbal rating pain scale, with an IQR of 5–9, while the median pain score at EDOU placement was 5 (IQR 2–7). Most patients’ pain was managed using hydromorphone (57.3%) and/or morphine (55.0%). Less than half of the patients in the EDOU required consultations for pain management, with 15.5% requiring pain service consultations and 30.0% requiring supportive care consultations (Table 3).

Significantly higher proportions of patients who were ultimately admitted to the hospital had received pain service or supportive care consultations compared to those who were ultimately discharged (Table 4). Specifically, only 10.2% of discharged patients had a pain service consultation, in contrast to 24.0% of admitted patients (*p* < 0.001). Similarly, 36.7% of patients requiring inpatient admission had a supportive care consultation, in contrast to 25.9% of discharged patients (*p* = 0.005).

Importantly, patients discharged from the EDOU after pain management had low 14-day and 30-day mortality rates (Table 4); only 0.3% of discharged patients died within 14 days of discharge (vs. 3.1% of admitted patients, *p* = 0.006), and 1.6% of discharged patients died within 30 days of discharge (vs. 7.9% of admitted patients, *p* < 0.001). Though 72 h ED revisit rates were higher in discharged patients compared to admitted patients, they remained low, with only 2.2% of the patients who were discharged after pain management in the EDOU revisiting the ED within 72 h (vs. 0.0% of admitted patients, *p* = 0.027, Table 4). Furthermore, patients who were admitted had significantly higher initial ED pain scores (*p* = 0.010), higher median pain scores during the ED stay (*p* < 0.001), and higher initial EDOU pain scores (*p* < 0.001) compared to the patients who were discharged (Table 5).

Univariate analysis showed that the initial ED triage pain score and the initial EDOU pain score were both associated with hospital admission after EDOU placement. In addition, pain/chronic pain consultation, palliative/supportive care consultation, and prescription of opioids prior to ED presentation were also associated with hospital admission (Table 5). Similarly, in the multivariable analysis, the initial EDOU pain score (OR = 1.12; 95% CI 1.06–1.19; *p* < 0.001), pain/chronic pain consultation (OR = 2.67; 95% CI 1.63–4.40; *p* < 0.001), palliative/supportive care consultation (OR = 2.04; 95% CI 1.37–3.04; *p* < 0.001), and prescription of opioids prior to ED presentation (OR = 1.52; 95% CI 1.06–2.19; *p* = 0.024) were all associated with hospital admission, after controlling for other clinical variables.

## 4. Discussion

The EDOU appears to be an appropriate venue for the management of uncontrolled cancer pain. Our results showed that most patients suffering from cancer pain who are placed in an EDOU for pain management can be safely discharged. Discharged patients have low ED revisit rates within 72 h as well as low 14-day and 30-day mortality rates after discharge from an EDOU. In fact, the 72 h rate of return found in this study (1.3%) is notably lower than what was observed in a study examining revisit rates in a general EDOU [12]. The low ED revisit and mortality rates in patients discharged from this EDOU after presenting with pain align with the overall low rate of adverse events experienced by all patients discharged from this EDOU. Interestingly, in a prior study, the ED revisit rate for patients discharged from this EDOU regardless of chief complaint was lower than that of patients discharged from its associated ED [13]. We also found that patients admitted to the hospital after their cancer pain was managed in an EDOU had a higher mortality rate compared to those who were discharged from the EDOU. This may be because the pain experienced by patients who ultimately required inpatient admission heralded poorer health and disease progression necessitating that admission.

Though the authors were unable to find another descriptive analysis of patients placed in an EDOU for pain as a primary concern, this study’s results were in line with several other publications exploring the characteristics associated with cancer-related pain. One such chart review of mostly gastrointestinal and lung cancer patients found that factors associated with more difficult-to-control pain included having a gastrointestinal tumor and severe baseline pain [14]. Similarly, our study showed that colorectal cancer was one of the most common types of cancer leading to placement in an EDOU for cancer pain management.

We also demonstrated that patients who were admitted to the hospital from the EDOU were more likely to have had supportive care or pain service consultations during their EDOU stay, indicating that their pain may have remained more intense or been more intractable than that experienced by patients who would eventually be discharged. Indeed, patients who were ultimately hospitalized had higher median pain scores during their ED visit and higher median pain scores during their EDOU stay compared to those who were discharged. Initial pain scores in both the ED and EDOU were also higher among patients who were eventually admitted to the hospital compared to those who were discharged. Multiple other studies similarly demonstrate that initial pain intensity is the most important domain in predicting successful pain management, with increased intensity being associated with a longer time to adequate pain control [15,16]. It is possible that patients who required supportive care or pain service consultations had more complex pain syndromes or that the interventions recommended by consultants required more time than would have been viable in an EDOU. Future studies could explore the nature and impact of the therapeutic interventions afforded to patients in the EDOU.

In a large study by Knudsen et al., initial pain control was an important predictor of improved pain relief in patients with breakthrough cancer pain [15], indicating that an EDOU, which in our study approximately housed the first day of a patient’s pain management, can serve as a tool for more ideal pain control. When analyzing the characteristics of patients suffering from cancer-related pain, our study agreed with studies by Coyne et al. and Knudsen et al., which found that pain was most often localized to the abdomen and back and that the most common types of cancer associated with persistent pain were gastrointestinal/colorectal, lung, and breast cancers [5,15].

Our study’s data reflected the wide range of cancers that patients presenting to an emergency department with pain are often diagnosed with and distinguished between various cancers in its analyses. Unlike a study by Wang et al., which showed that gastrointestinal cancer was an independent risk factor for poor treatment outcomes in 3 days, our study found that the type of cancer a patient was diagnosed with did not have a significant impact on whether they were admitted or discharged from an EDOU, a decision generally made in approximately 1 day after placement in our EDOU (Table 4). Whereas Wang et al. found that gastrointestinal cancer-related breakthrough pain took a median of 3 days to control and lung cancer-related breakthrough pain took a median of 2 days to control, our study found no significant predictors of admission with regards to the type of cancer a patient had (Table 4) [14]. In our whole study population, the median length of stay for patients who were placed in our EDOU was just 26 h, with an IQR of 18–42 h (Table 6).

This difference in findings may be related to differences between the populations analyzed in our study vs. the study conducted by Wang et al. While Wang et al. reviewed the charts of patients with an inpatient hospitalization for cancer pain, our study reviewed the 13.6% of patients presenting to the ED with pain as a primary concern who were placed in an EDOU [14]. Though these findings may reflect differences produced through EDOU management, it is important to consider that our study reviewed an already stratified population of patients whose pain was deemed manageable in an EDOU setting as opposed to requiring inpatient hospitalization. Future studies comparing the characteristics that lead to inpatient admission, discharge, and EDOU placement from an ED may be helpful in elucidating the differences in outcomes between the Wang et al. study and ours.

Pain-related ED visits have historically been and continue to be common among patients with breakthrough oncologic pain [4,17]. As early as 2002, the cost of care delivered during pain-related hospitalizations for these patients was estimated to equal USD 1.7 million per patient each year, and the per capita costs of cancer-related care have only grown since then [17,18]. For other types of pain, such as chest pain, EDOUs have proven instrumental in the initial management and risk stratification of patients presenting to an ED [19]. In this capacity, EDOUs have been estimated to save the healthcare system USD 3.1 billion a year when used to care for lower-risk patients who are not appropriate for discharge [20].

Many patients in our study population who were placed in an EDOU for pain management required this placement despite outpatient management, as almost one-third of the patients analyzed were already seeing supportive care and pain service specialists outside of the hospital (Table 1). Given the fact that most patients seen for oncologic pain in our EDOU were discharged successfully with low 72-hour ED revisit and 30-day mortality rates, it appears that observation medicine may safely reduce the number of unnecessary admissions associated with emergent presentations of cancer pain. As a high percentage of cancer patients in the United States continue to require the services of an ED in the management of their oncologic pain, EDOUs have the potential to decrease the financial toxicity of cancer care and protect cancer patients from the hospital-acquired infections and recurrent complications that they are susceptible to when hospitalized [21]. Though more studies are necessary to determine the exact cost savings that EDOUs can bring in cancer care, our study lends credence to the idea that EDOUs may serve as a cost-saving tool for the provision of appropriate pain management for our patients with cancer-related pain.

## 5. Limitations

As there are only two cancer-specific EDOUs in the world, our study was limited to a single-center trial. However, as our study site routinely treats a large number of patients with a wide variety of cancers who have received a diverse range of treatments, the external validity of our study’s findings may only be minimally affected by the single-center cohort we analyzed. Our study bears the same limitations that other retrospective chart reviews bear, with the data being limited by real-time record-keeping practices. For example, we were not able to confirm the performance status of every patient in the study, instead relying on a patient’s need for supportive care or chronic pain service involvement to infer the impact of their disease on quality of life. However, our data about morbidity and mortality were robust, as patients presenting to our ED were likely to also be followed by MD Anderson Cancer Center specialists in the outpatient setting. This is supported by the fact that many of the patients in our study were followed by our supportive care and pain services prior to presentation.

The scope of this study was defined by data from patients who were already placed in an EDOU, limiting our analyses to patients who were selected by treating physicians for observation as opposed to admission or discharge. It is, thereby, outside this study’s scope to delineate if patients with specific cancers, comorbidities, or other characteristics are more likely to be directly admitted or observed upon initial presentation to the ED. Further research comparing these factors for patients who were directly admitted to the hospital from the ED when presenting with pain with the same factors for those who were placed in an EDOU may help inform predictive decision-making for physicians and hospital systems deciding which patients with oncologic pain should be placed in an EDOU. Social factors were also outside the scope of this study, as socioeconomic well-being and a patient’s ability to follow up were not parsed out in our analysis. Further study on the effect these factors have on cancer-related care is important because a patient’s socioeconomic well-being may affect their ability to get outpatient follow-up and advanced care. This, in turn may have an effect on the rate of ED revisits within 72 h of discharge from an EDOU as well as the 30-day mortality rate in patients with fewer financial resources.

## 6. Conclusions

EDOUs are capable of successfully managing a subsegment of patients presenting to an emergency department for acute oncologic pain, with low 72-hour ED revisit and 30-day mortality rates. Though the most frequent types of cancer among patients presenting with pain are gastrointestinal, breast, and lung cancers, the type of cancer a patient has is not significantly associated with the patient’s disposition from an EDOU. A majority of patients cared for in an EDOU are successfully discharged, with hospitalized patients more often requiring supportive care or pain service consultations prior to admission. Our findings suggest that EDOUs offer a more appropriate care environment than immediate admission or discharge for patients who require significant cancer pain management but may not require inpatient hospitalization for their therapy.

## Figures and Tables

**Figure 1 cancers-14-05871-f001:**
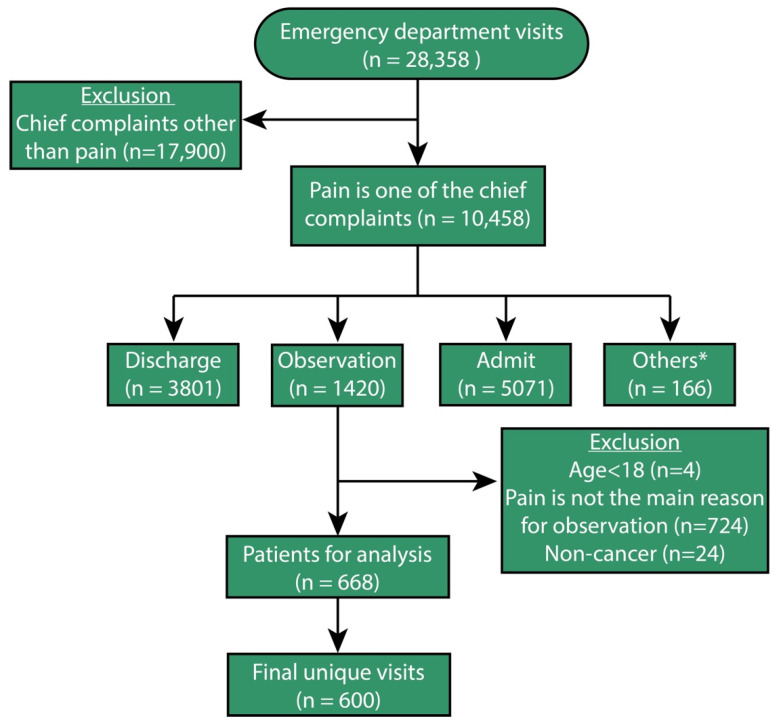
Flow diagram showing the patient selection steps. * Including patients who passed away, who were transferred, and who left against medical advice.

**Table 1 cancers-14-05871-t001:** Characteristics of cancer patients placed in an emergency department observation unit with pain.

Characteristics	No. (%)
Total	600
Age, median (IQR), years	59 (48–68)
Sex	
Female	365 (60.8)
Male	235 (39.2)
Race/ethnicity	
White	351 (58.5)
Black	120 (20.0)
Hispanic/Latino	79 (13.2)
Asian	24 (4.0)
Others	26 (4.3)
Primary cancer type	
Breast	78 (13.0)
Colorectal	56 (9.3)
Lung	48 (8.0)
Sarcoma	43 (7.2)
Endometrial and cervical	40 (6.7)
Lymphoma	38 (6.3)
Pancreas	36 (6.0)
Gastroesophageal	32 (5.3)
Kidney	30 (5.0)
Head and neck	26 (4.3)
Male genital	23 (3.8)
Multiple myeloma	21 (3.5)
Hepatobiliary	18 (3.0)
Ovarian and fallopian tube	15 (2.5)
Others	96 (16.0)
Secondary cancer	
No	584 (97.3)
Yes	16 (2.7)
Cancer treatment prior to ED visit	
Chemotherapy within 30 days	
No	420 (70.0)
Yes	180 (30.0)
Radiotherapy within 30 days	
No	419 (69.8)
Yes	181 (30.2)
Immunotherapy within 1 year	
No	521 (86.8)
Yes	79 (13.2)
Comorbidities	
Hypertension	281 (46.8)
Heart failure	39 (6.5)
Myocardial infarction	35 (5.8)
COPD	40 (6.7)
Diabetes	144 (24.0)
Renal failure	126 (21.0)
Established with pain/chronic pain services	
No	504 (84.0)
Yes	96 (16.0)
Established with palliative/supportive care services	
No	503 (83.8)
Yes	97 (16.2)
Opioid prescription within 45 days prior to presentation	
No	261 (43.5)
Yes	339 (56.5)

Abbreviations: IQR, interquartile range; COPD, chronic obstructive pulmonary disease; ED, emergency department.

**Table 2 cancers-14-05871-t002:** Characteristics of cancer patients placed in an emergency department observation unit with pain.

Characteristics	No. (%)
Location/type of pain *	
Abdominal pain	174 (29.0)
Back pain	104 (17.3)
Extremities pain	95 (15.8)
Chest pain	55 (9.2)
Urinary and genital pain	27 (4.8)
Neck pain	12 (2.0)
Generalized pain	5 (0.8)
Others/miscellaneous	165 (27.5)
Associated nausea and/or vomiting	
No	460 (76.7)
Yes	140 (23.3)
Initial ED triage pain score, median (IQR)	7 (5–9)
Initial EDOU pain score, median (IQR)	5 (2–7)

Abbreviation: ED, emergency department; IQR, interquartile range; EDOU, emergency department observation unit. * Some patients had more than one type of pain during their visit; therefore, numbers do not add up to 100%.

**Table 3 cancers-14-05871-t003:** Management and outcomes of cancer patients with pain placed in an emergency department observation unit.

Management	No. (%)
Consultation	
Pain/chronic pain	93 (15.5)
Palliative/supportive care	180 (30.0)
Type of pain medication *	
Hydromorphone	344 (57.3)
Morphine	330 (55.0)
Oxycodone	111 (18.5)
Combination opioid	123 (18.4)
Tramadol	79 (13.2)
Fentanyl	73 (12.2)
Methadone	49 (8.2)
Percocet	17 (2.8)
Others	12 (2.0)

* Some patients had more than one type of pain medication during their visit; therefore, numbers do not add up to 100%.

**Table 4 cancers-14-05871-t004:** Clinical characteristics and outcomes of patients who were admitted to the emergency department observation unit for pain stratified by disposition.

Variable	Disposition		*p*
	Admission	Discharge	
Total	229	371	
Age, median (IQR), years	59 (48–69)	59 (47–67)	0.890
Sex			0.691
Female	137 (59.8)	228 (61.5)	
Male	92 (40.2)	143 (38.5)	
Race/ethnicity			0.781
White	132 (57.6)	219 (59.0)	
Black	45 (19.7)	75 (20.2)	
Hispanic/Latino	29 (12.7)	50 (13.5)	
Asian	12 (5.2)	12 (3.2)	
Others	11 (4.8)	15 (4.0)	
Cancer type			0.089
Brain and spinal cord	3 (1.3)	5 (1.3)	
Breast	31 (13.5)	47 (12.7)	
Colorectal	23 (10.0)	33 (8.9)	
Endometrial and cervical	17 (7.4)	23 (6.2)	
Gastroesophageal	13 (5.7)	19 (5.1)	
Head and neck	8 (3.5)	18 (4.9)	
Hepatobiliary	3 (1.3)	15 (4.0)	
Kidney	13 (5.7)	17 (4.6)	
Leukemia	1 (0.4)	4 (1.1)	
Lung	17 (7.4)	31 (8.4)	
Lymphoma	14 (6.1)	24 (6.5)	
Male genital	3 (1.3)	20 (5.4)	
Melanoma	3 (1.3)	10 (2.7)	
Multiple myeloma	11 (4.8)	10 (2.7)	
Neuroendocrine tumors	1 (0.4)	4 (1.1)	
Others	22 (9.6)	15 (4.0)	
Ovarian and fallopian tube	2 (0.9)	13 (3.5)	
Pancreas	13 (5.7)	23 (6.2)	
Sarcoma	21 (9.2)	22 (5.9)	
Skin	2 (0.9)	5 (1.3)	
Small intestine	0 (0.0)	1 (0.3)	
Thyroid	4 (1.7)	3 (0.8)	
Urinary bladder and ureter	4 (1.7)	9 (2.4)	
Chemotherapy within 30 days			0.111
No	169 (73.8)	251 (67.7)	
Yes	60 (26.2)	120 (32.3)	
Radiotherapy within 30 days			0.042
No	171 (74.7)	248 (66.8)	
Yes	58 (25.3)	123 (33.2)	
Immunotherapy within 1 year			0.970
No	199 (86.9)	322 (86.8)	
Yes	30 (13.1)	49 (13.2)	
Hypertension			
No	112 (48.9)	207 (55.8)	0.101
Yes	117 (51.1)	164 (44.2)	
Heart failure			
No	209 (91.3)	352 (94.9)	0.081
Yes	20 (8.7)	19 (5.1)	
Myocardial infarction			
No	212 (92.6)	353 (95.1)	0.192
Yes	17 (7.4)	18 (4.9)	
COPD			
No	207 (90.4)	353 (95.1)	0.023
Yes	22 (9.6)	18 (4.9)	
Diabetes			
No	167 (72.9)	289 (77.9)	0.166
Yes	62 (27.1)	82 (22.1)	
Renal failure			
No	167 (72.9)	307 (82.7)	0.004
Yes	62 (27.1)	64 (17.3)	
Length of stay, median (IQR), hours	26 (18, 41)	27 (19, 44)	0.244
Pain/chronic pain consultation			<0.001
No	174 (76.0)	333 (89.8)	
Yes	55 (24.0)	38 (10.2)	
Palliative/supportive care consultation			0.005
No	145 (63.3)	275 (74.1)	
Yes	84 (36.7)	96 (25.9)	
ED revisit within 72 h			
No	229 (100.0)	363 (97.8)	0.025
Yes	0 (0.0)	8 (2.2)	
Death within 14 days			
No	222 (96.9)	370 (99.7)	0.004
Yes	7 (3.1)	1 (0.3)	
Death within 30 days			
No	211 (92.1)	365 (98.4)	<0.001
Yes	18 (7.9)	6 (1.6)	

COPD, chronic obstructive pulmonary disease; ED, emergency department; IQR, interquartile range.

**Table 5 cancers-14-05871-t005:** Univariate and multivariable logistic regression analyses of the association of clinical variables with hospital admission after emergency department observation unit placement.

Variable	Univariate		Multivariable	
	OR (95% CI)	*p*	OR (95% CI)	*p*
Age	1.00 (0.99–1.01)	0.977	1.00 (0.99–1.02)	0.500
Sex				
Female	Reference			
Male	1.07 (0.76–1.50)	0.691	-	-
Race/ethnicity				
Non-White	Reference			
White	0.89 (0.53–1.53)	0.669	-	-
Cancer type				
Hematologic	Reference			
Solid	0.89 (0.53–1.53)	0.669	0.97 (0.54–1.77)	0.924
Established with pain/chronic pain services	1.46 (0.94–2.27)	0.093		
Established with palliative/supportive care services	1.17 (0.75–1.81)	0.497		
Opioid prescription within 45 days prior to presentation	1.77 (1.27–2.50)	<0.001	1.52 (1.06–2.19)	0.024
Chemotherapy	0.74 (0.51–1.07)	0.111	-	-
Radiotherapy	0.68 (0.47–0.98)	0.043	0.68 (0.45–1.00)	0.054
Immunotherapy	0.99 (0.60–1.60)	0.970	-	-
Pain/chronic pain consultation	2.77 (1.77–4.38)	<0.001	2.67 (1.63–4.40)	<0.001
Palliative/supportive care consultation	1.66 (1.16–2.37)	0.005	2.04 (1.37–3.04)	<0.001
Location/type of pain				
Abdominal pain	0.77 (0.53–1.11)	0.170	-	-
Back pain	1.02 (0.65–1.56)	0.946	-	-
Extremities pain	0.99 (0.62–1.54)	0.953	-	-
Chest pain	0.64 (0.34–1.15)	0.149	-	-
Urinary and genital pain	0.74 (0.30–1.66)	0.477	-	-
Initial ED triage pain score	1.07 (1.01–1.14)	0.015	-	-
Initial EDOU pain score	1.16 (1.09–1.22)	<0.001	1.12 (1.06–1.19)	<0.001
Pain difference between initial ED triage and initial EDOU pain score	1.06 (1.02–1.11)	0.009	-	-
Nausea and/or vomiting	0.74 (0.49–1.10)	0.140	0.89 (0.58–1.35)	0.577

Abbreviations: ED, emergency department; EDOU, emergency department observation unit; OR, odds ratio.

**Table 6 cancers-14-05871-t006:** Disposition and outcomes of cancer patients with pain placed in an emergency department observation unit.

Outcome	No. (%)
Length of stay, median (IQR), hours	26 (18–42)
Disposition	
Admission	229 (38.2)
Discharge	371 (61.8)
ED revisit within 72 h	
No	592 (98.7)
Yes	8 (1.3)
Death within 14 days	
No	592 (98.7)
Yes	8 (1.3)
Death within 30 days	
No	576 (96.0)
Yes	24 (4.0)

IQR, interquartile range; ED, emergency department.

## Data Availability

The data presented in this study are available within the article in the form of tables.

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
