# Peer review of "Characteristics and Outcomes of Patients with Cancer Pain Placed in an Emergency Department Observation Unit"

_cancers, 2022, doi:10.3390/cancers14235871_

Round 1

Reviewer 1 Report

Major comments

The emergency department observation unit (EDOU) affords patients access to a hospital’s resources without hospitalization. To define the role of an EDOU in the management of cancer pain, the authors conducted a retrospective study analyzing patients placed in an EDOU with uncontrolled cancer pain over a year. Patient characteristics were summarized using descriptive statistics and predictors of disposition from the EDOU were identified with univariate and multivariate analyses. Most patients were discharged home, and discharged patients had low 72-hour revisit and 30-day mortality rates. Significant predictors of hospitalization were initial EDOU pain score (odds ratio [OR]=1.12; 95% CI 1.06-1.19; P<0.001) and supportive care (OR-2.04; 95% CI 1.37-3.04; P<0.001) or pain service (OR=2.67; 95% CI 1.63-4.40; P<0.001) consultations. The authors concluded that EDOUs appear to be appropriate venues to care for a subsegment of patients presenting to an emergency department with cancer pain, with patients receiving safe care as well as appropriate consultation and admission when indicated.

              This referee is sure that this study was a sound retrospective study analyzing patients placed in an EDOU with uncontrolled cancer pain about the disposition from the EDOU. The manuscript was also written well.

Author Response

Thank you so much for your comments and for the time you spent reviewing our article. We hope that you enjoyed reading it!

Reviewer 2 Report

The authors are all from the Emergency Medicine Department at MD Anderson Hospital.  The decision to admit patients to the observation unit seems to have been done from the Emergency Medicine perspective and the data in the article provides information about the population who are admitted to the Observation Unit.  If the overall goal of the study is to give emergency medicine physicians information about individuals with cancer and with pain in an observation unit, this goal has been achieved. 

However, there are several questions raised if a different perspective is used which might also be useful to Emergency Medicine physicians, in general, but also oncologists and palliative care specialists.  For example, what was the decision path to put these individuals in the observation unit as opposed to the hospital or discharged directly from the ED.  Were logistics—bed availability, e.g., playing a role.  Was the patient’s primary oncologist or oncology team consulted?  Would a comparison of the 5071 patients admitted directly to hospital using the same variables be of value to all groups of physicians? 

As with all retrospective studies, there are inherent weaknesses in these observations:

1.       In particular, the performance status of these patients and the disease stage are not indicated.  The higher rate of admission for those with supportive care consults and a near doubling of those who had such a consult from baseline suggests that a significant proportion of the patients have more advanced disease, or a worse performance status, but this is an inference.  

2.       The authors do not note if some of these patients were seen in the ED for inadequate pain control.  The number on prior pain medication is not stated.

3.       Another way to look at these individual’s pain would be to look at the site of pain—i.e., bone, visceral, etc.  The large number of patients with breast cancer or lung cancer might be accounted for by a common site of metastatic spread (if in fact, many of these patients had systemic disease as opposed to loco-regional disease). 

4.       The authors compare their findings to those of Wang (ref 14)   This population was composed of Chinese individuals admitted to a hospital in China with uncontrolled pain—the most likely source of pain from a gastrointestinal cancer standpoint in such a population would be cancer of stomach/esophagus/gastroesophageal junction and is thus less likely to be comparable to a U.S. population where colorectal cancer is more common. 

5.       There are several other cancers at a low frequency in this study—prostate cancer, pancreatic cancer—which also suggest that the population as a whole is highly selected; but it is not possible to discern this from the data presented which is both retrospective and post hoc the decision to place the patient in an Observation unit. 

While some of these issues may be beyond this data set to sort out, some consideration by the authors would be appreciated. 

Finally, the references are numbered twice—this duplication needs to be corrected.

Author Response

POINT 1:  For example, what was the decision path to put these individuals in the observation unit as opposed to the hospital or discharged directly from the ED.  Were logistics—bed availability, e.g., playing a role.  Was the patient’s primary oncologist or oncology team consulted?  

RESPONSE 1: We thank this reviewer for this question and have included more information about the decision tree for observation in the patient population studied. The new text added is as follows between lines 78-87: "Patients were placed in the EDOU during the study’s timeframe only when medically necessary. Patients were placed in observation as opposed to hospitalized inpatient based on the two-midnight rule for Medicare patients and Millman care guidelines for managed care or commercial payers. Screening using this criteria was achieved at the time of treating physician documentation with the assistance of case managers, as was standard during the study period at The University of Texas MD Anderson Cancer Center. Primary oncologists were notified about all placements in the EDOU and thereby had an opportunity to change their patients’ plan of care. When the EDOU no longer had bed capacity, patients were placed in observation on their primary service – these patients were not included in this study’s analysis."

POINT 2: Would a comparison of the 5071 patients admitted directly to hospital using the same variables be of value to all groups of physicians? 

RESPONSE 2: In lines 313-316, we have also included text about the comparison suggested between patients admitted from the ED versus those who were placed in an EDOU. Though it does not fall within the scope of this study, it is an excellent opportunity for further research. The new text is as follows: "Further research comparing these factors for patients who were directly admitted to the hospital from the ED when presenting with pain with those who were placed in an EDOU may help inform predictive decision making for physicians and hospital systems deciding which patients with oncologic pain should be placed in an EDOU." 

POINT 3: In particular, the performance status of these patients and the disease stage are not indicated.  The higher rate of admission for those with supportive care consults and a near doubling of those who had such a consult from baseline suggests that a significant proportion of the patients have more advanced disease, or a worse performance status, but this is an inference.  

RESPONSE 3: This is an important limitation inherent in the retrospective design of our study and we have included text addressing it between lines 300-303. The new text included reads: "For example, we were not able to confirm the performance status of every patient in the study, instead relying on patients’ need for supportive care or chronic pain service involvement to infer the impact of their disease on quality of life."

POINT 4: The authors do not note if some of these patients were seen in the ED for inadequate pain control.  The number on prior pain medication is not stated.

RESPONSE 4: As noted in lines 119-121, the patients in the study were presenting to the ED with pain. Further, the number of patients on opioid pain medications prior to presentation is noted at the bottom of Table 1. In Table 5 and lines 189-197, it is seen and discussed that a prescription for opioids before ED presentation was associated with admission from the EDOU.

POINT 5: Another way to look at these individual’s pain would be to look at the site of pain—i.e., bone, visceral, etc.  The large number of patients with breast cancer or lung cancer might be accounted for by a common site of metastatic spread (if in fact, many of these patients had systemic disease as opposed to loco-regional disease). 

RESPONSE 5: The location of pain is explored in Table 2 and discussed in the text in lines 147-149. In table 5, it is seen that the location of pain was not significantly associated with admission to the hospital from the EDOU. Localization of pain was also discussed in lines 240-244 and with this feedback, we have referenced another article in our bibliography to enhance the text discussing localization of pain. These lines now read: "When analyzing the characteristics of patients suffering from cancer-related pain, our study agreed with studies by Coyne et al and Knudsen et al, which found that pain was most often localized to the abdomen and back and that the most common types of cancer associated with persistent pain were gastrointestinal/colorectal, lung, and breast cancers [5, 15]."

POINT 6: The authors compare their findings to those of Wang (ref 14)   This population was composed of Chinese individuals admitted to a hospital in China with uncontrolled pain—the most likely source of pain from a gastrointestinal cancer standpoint in such a population would be cancer of stomach/esophagus/gastroesophageal junction and is thus less likely to be comparable to a U.S. population where colorectal cancer is more common. 

RESPONSE 6: We have already accounted for this feedback by differentiating between gastroesophageal cancers and colorectal cancers in our analysis, specifically in Tables 1 & 4. As discussed in lines 248-250, where Wang et al is referenced, we found that "the type of cancer a patient was diagnosed with did not have a significant impact on whether they were admitted or discharged from an EDOU" despite this differentiation.  We have included text to highlight the range of cancers included in this study in lines 245-247, referencing another article in our bibliography. The new text reads "Our study’s data reflected the wide range of cancers that patients presenting to an emergency department with pain are often diagnosed with and distinguished between various cancers in its analyses (Tables 1 & 4)." 

POINT 7:  There are several other cancers at a low frequency in this study—prostate cancer, pancreatic cancer—which also suggest that the population as a whole is highly selected; but it is not possible to discern this from the data presented which is both retrospective and post hoc the decision to place the patient in an Observation unit. 

RESPONSE 7: As astutely recognized by this reviewer, the data is limited to patients selected for observation by a treating physician. In order to address this feedback, we have included the following text in lines 308-312: "The scope of this study was defined by data from patients who were already placed in an EDOU, limiting our analyses to patients who were selected by treating physicians for observation as opposed to admission or discharge. It is thereby outside this study’s scope to delineate if patients with specific cancers, comorbidities, or other characteristics are more likely to be directly admitted or observed upon initial presentation to the ED."

POINT 8: Finally, the references are numbered twice—this duplication needs to be corrected.

RESPONSE 8: The reference numbers appear to have been erroneously duplicated in the submission process, these have been corrected, with the duplicates being deleted.